# Child Developmental Disabilities, Caregivers’ Role in Kenya and Its Implications on Global Migration

**DOI:** 10.3390/ijerph16061010

**Published:** 2019-03-20

**Authors:** Jemaiyo Chabeda-Barthe, Timothy Wambua, Wangui Lydia Chege, Dan Hwaga, Timothy Gakuo, Gladys Chepkemoi Rotich

**Affiliations:** 1Institute of Social Anthropology, University of Bern, Lerchenweg 36, CH-3000 Bern 9, Switzerland; 2Kenya Institute of Special Education, P.O. Box 48413, Nairobi 00100, Kenya; wambuat@kise.ac.ke (T.W.); chegel@kise.ac.ke (W.L.C.); hwagad@kise.ac.ke (D.H.); gakuot@kise.ac.ke (T.G.); rotichg@kise.ac.ke (G.C.R.)

**Keywords:** migration, developmental disability, children, inclusion

## Abstract

*Background*: This paper is a summary of the findings from an ethnographic study on child developmental disabilities conducted partly in Nairobi and Kiambu Counties in Kenya. *Methods*: Quantitative and qualitative data collection methods were applied for the period between mid August and mid November 2018. The study was conducted through the Kenya Institute of Special Education (KISE) situated in Nairobi County. *Results*: There are parents who are willing to migrate in search of better education and healthcare options for their children who have developmental disabilities (DDs). However, there are also government reforms taking place in the field of disability that may help to support the caregiving role for children with special needs. The challenges, bargaining position and power play between parents or guardians and other actors implicated in the debates on inclusion and integration of persons with developmental disabilities in Kenya has been brought to the forefront. *Conclusions*: In Kenya, more needs to be done to change the attitude towards disability from the medical and moral (religious/cultural) models to an approach leaning towards the social model, so that developmental disabilities are not viewed negatively.

## 1. Introduction

The World Migration report for 2018 estimates that there are approximately 244 million migrants worldwide [1]. In Europe, the influx of migrants from the African continent has sharply increased since the political uprisings also known as the “Arab spring” in December 18, 2010, which occurred in North African countries [2]. What began as the need to migrate due to displacement because of the civil war has gradually developed into economic migration with many sub Saharan Africans using the North African countries as a conduit route [3]. In countries such as Djibouti, Kenya, Uganda and Tanzania which are located in the horn of Africa, some of these economic migration patterns are voluntary whilst some are as a result of forced migration or smuggling.

It is for this reason that the European Union has been conducting rigorous trainings in the region about the dangers of smuggling and trafficking of humans. The Better Migration Management Project implemented through the International Office of Migration in Nairobi, has been training officials from the police investigative departments, witness protection agencies, refugee councils and human rights commissions in Djibouti, Kenya, Uganda, Ethiopia and Tanzania.

The relationship between disability, poverty and migration in the horn of Africa region has been brought to the forefront through children. As argued by Groce et al. ([4], p. 1493): “The international development community is beginning to recognize that people with disabilities constitute among the poorest and most vulnerable of all groups, and thus must be a core issue in development policies and programmes). In East Africa, there are reports about child smugglers bringing physically handicapped children from Tanzania [5], to beg on the streets of Nairobi [6]. Other reports, demonstrate that there exists the abduction of albino children from Kenya or Uganda who are taken for sacrificial rituals in countries such as Tanzania [7]. Whilst the researcher found these aforementioned topics alarming, she chose to focus on developmental disabilities and the caregivers’ social and economic role. This article will demonstrate why there needs to be a societal change in the way child developmental disability is understood [8], and handled in a developing country such as Kenya.

## 2. Background

According to the 2008 United Nations Convention on the Rights of Persons with Disabilities (CRPD), disability is a concept that is based both on the individuals’ impairment and social context. The report by the World Health Organization (WHO) (2015) [9], defines disability as “reflecting the interaction between features of a person’s body and features of the society in which he or she lives” (Disabilities World Health Organization, 2015). Intellectual disability is defined by WHO as: “a significantly reduced ability to understand new or complex information and to learn and apply new skills (impaired intelligence). This results in a reduced ability to cope independently (impaired social functioning), and begins before adulthood, with a lasting effect on development” (Definition: intellectual disability, 2015 [9]). In this article, the term ‘developmental disability’ or DD, will be used to represent both intellectual disabilities and learning disorders.

Research on disability is gaining ground in Africa [10]. In Kenya, the national special needs education policy framework of 2009 [11], outlines twenty two categories of disabilities and special needs. These include children who have hearing, visual and physical impairments. In addition, cerebral palsy, epilepsy, Down syndrome, autism, emotional and behavioral disorders, learning disability, speech and language disorders, multiple disabilities and albinism. At national level, the term ‘mental disability’ has been shunned because it stigmatizes the child and the parent through affiliate stigma. The new government policies on disability embrace the terminology ‘intellectual disability’. However, the discriminatory mentality is historical and is still entrenched at grass root level. This is reflected in the naming of schools or rehabilitation centres for persons with disabilities.

In March 2018, a national survey on children with disabilities and special needs in education [12], was conducted by the Kenya Institute of Special Education (KISE). This KISE survey targeted an age group between 3 to 20 years. A total of 7609 children were reached, 865 had disabilities, which translates to a prevalence of 11.4%. Of the 865, 51.2% were boys while 48.8% were girls. Intellectual disabilities were 2.5 percent compared to physical disability which was 3.0 % and visual impairment the highest at 3.1%. These statistics show that children with disabilities form a significant number in Kenya and equally significant are those parents who provide care to them.

For this study, which is funded by the Swiss government, the aim is to look at how the developmental disability of a child can be a factor that may motivate parents to migrate to a country with presumed better education and healthcare facilities for the child. The findings will have policy implications on migration and disability in the region because; *firstly*, it has been noted that, on one hand, the number of children with detectable developmental disabilities such as Down syndrome, is rising in developing countries [13]. Therefore refugees or economic migrants may already have a child with special needs when migrating to a developed country such as Switzerland. On the other hand, detectable DDs are diminishing in developed countries because legal abortions and pre-natal testing are often options given to parents in such countries [14]. In most developing countries such as those in Africa, even where pre-natal testing may be done, there is still the element of strong religious beliefs and the absence of laws that support abortions.

*Secondly*, the findings will have implications on policymaking because, there are reports of migrants with professional background being denied visa on the grounds of their child’s special needs. In Australia, a migrant couple from Bangladesh who are both doctors and teach at Monash University, were denied permanent residence because of their daughter who has autism [15]. Until recently Canadian immigration laws were also discriminatory towards parents of children with special needs [16]. While in the United Kingdom, a report shows that migrants whose children have special needs are sometimes denied access to funds that are set aside for persons with disabilities [17]. In addition, migrants with disabilities often face challenges when trying to access healthcare and social benefits associated with disability [18].

*Thirdly*, this study is important because it is the first part of a broader project. The second part of the project is titled “Navigating the caregiving role between sports inclusion and developmental disabilities of children and youth: Case of migrant parents in Jura Bernois, Switzerland.” This project will eventually look at the situation of migrant families living in Switzerland, who in particular have a child or children with a DD. This study will contribute to the consequent broader study by showing how affiliate stigma is experienced by such parents. In addition, how attitudes about disability can be maintained and reproduced by migrant communities even when they have settled in a new country of residence such as Switzerland. Finally, it may demonstrate how such reproduced attitudes towards disability can negatively affect the social integration of the child into the society. This has implications on social integration policies for migrant communities in Switzerland and Europe in general. Sports or sporting activities are targeted for social integration in Switzerland [19].

In Switzerland, Hedderich and Lescow (2015) have highlighted the extra effort put into parenting and caregiving by parents of migrant background with at least one child who has a disability [20]. Other studies have looked at mental health and migration from the causal approach; that migration may cause mental health issues. It has been established by Chatzidiakou et al. (2016) that the traumatic experiences that refugees undergo during migration may lead to them to develop psychosocial issues in the new country of destination. They highlight the acute psychiatric health problems experienced by migrants living in Switzerland [21].

This article will examine the environmental barriers that children with DDs face and how these may cause parents to want to migrate to developed countries. Firstly, using the moral model of disability, the article will show how discrimination of children with DDs occurs within the context of educational facilities and how this may be an incentive to migrate. This will be followed by a snapshot on how the medical model of disability affects children with a DD. Thirdly, the human rights model will be used to show how the Kenyan government is reforming laws on disability through the national council of persons with disabilities (NCPWD). Fourth, the factors above will be used to demonstrate why some parents may choose to migrate. Lastly, there will be some discussions and the conclusion.

## 3. Theoretical Framework

This article incorporates the sustainable development goals (SDG) 3, target 3.4 which is partially about promoting mental health and well-being. This is examined using the caregivers’ theory and theories on social integration. On one hand, the caregivers’ theory show that parents with special needs children have an extra role to play as compared to those with typical children [22]. On the other hand, the theories on social integration differentiate four dimensions: social contacts and interactions, social relationships and friendships, social perception and social acceptance as factors that may influence the inclusion or lack thereof, of persons with developmental disabilities into the society [23]. In addition, studies have shown that beyond public and self-stigma, stigma can also impact family members. Only scant research has examined the internalized aspects of stigma, known as affiliate stigma, among family caregivers of individuals with disabilities [24]. According to Mak and Cheung (2008), affiliate stigma may contribute to the element of social exclusion which this study has explored [25]. The factors influencing the caregiving process as highlighted by King et al. (1999), are adapted [26].

In many developing countries such as Kenya, persons with DDs do not have the same rights as other people. Retief and Letšosa (2018) attribute this to the historical use of both the medical model and the moral (religious) model rather than the social model of disability [27]. On one hand, Thomas and Woods (2003), show that the medical model of disability approaches disability as a disease thus creating a need to seek medical treatment to find solutions to cure the “disease” ([27], p. 3). Furthermore, Carlson (2010), argues that through the medical approach, disability is regarded as ‘a personal tragedy for both the individual and his/her family, something to be prevented and, if possible, ‘treated’ ([27], p. 3). For the moral model (religious), disability is regarded as a punishment from God for a particular sin or sins that may have been committed by the person with a disability or by his or her parents ([27], p. 2). In medical and moral models, disability is seen as a burden. On the other hand, the social model approach uses the term ‘disabled people’ because it reflects the societal oppression that people with special needs are faced with every day ([27], p. 4). According to Purtell (2013), ‘[D]isabled people are people who are “disabled” by the society they live in and by the impact of society’s structures and attitudes’ ([27], p. 4).

## 4. Mixed Methods Design, Data Collection and Analysis

This is a mixed methods ethnographic study [28]. It is exploring the relational understanding of caregiving by parents of children with a DD, and how this has implications on global migration. Data was collected using four methods; survey, interviews, focus group discussions and participant observation. The researcher from the University of Bern was able to mobilize the target sample within a short timeframe because she is a member of T21 virtual support group for families whose children have DDs. In addition, being Kenyan and understanding the local language put her at an advantage as compared to say other non-Kenyan anthropologists. Her approach showed both sensitivity and detachment when designing the tools for data collection. Nairobi and Kiambu counties were selected because of the high concentration of special needs education facilities in the area as compared to other counties. Purposive and convenience sampling were employed. The communities were selected based on *firstly*, having a child with a developmental disability. A *second* criterion was the social economic status of parent. *Thirdly*, was their willingness to participate in the study and their geographical accessibility in Nairobi and Kiambu counties.

### 4.1. Survey

A total of 90 questionnaires were filled out by parents from both low and high social economic statuses and whose children have a form of DD (mainly focusing on; Down syndrome, autism, cerebral palsy and epilepsy). Questionnaires were administered manually to parents from low social economic status (SES) at the Kenya institute of special education (KISE). The responses were also collected manually. A web link was sent out to members of an organization called T21 families, through email and the *whatsup* group. The T21 families Community Based Organization was founded in 2018 by a mother of a child with Down syndrome based in Nairobi the capital city. It offers online advice on medical camps and rehabilitation services. Most of these parents are from high SES. Their responses were collected electronically. There were 51 responses from high SES and 39 responses from the low SES.

### 4.2. Interviews

15 interviews were conducted for this research. The interview schedules were varied according to the background of the interviewee. Interviews with officials from two Non-Governmental Organizations; Down Syndrome Society of Kenya (DSSK) and Leonard Cheshire International, focused on why inclusiveness is top on the agenda for projects on disability in Kenya. Interviews with three doctors from one of the following hospitals; Mathari, Kenyatta and Mater Misericordiae were about the role of the historical medical model of disability and the cost implication of health interventions associated with DDs. Interviews with special needs education practitioners from Saint Patrick’s School for the mentally handicapped children, Model Inclusive Pre-primary (situated with Kenya Institute of Special Education), and Maria Magdalena sheltered workshops for persons with disabilities centered on the challenges of running a public facility for persons with DDs.

The interviews with parents whose children are enrolled at Mirema private school (with a special needs section) was centered on the linkages between schools and private sector in creating job opportunities for persons with disabilities. Lastly, the interviews with government representatives at Kenya Institute of Special education and Kenya Institute of Curriculum Development centered on the national status of developmental disabilities and the intended policy reforms for the future. Some of the aforementioned institutions are highlighted in Figure 1 below.

### 4.3. Focus Group Discussions

A total of five focus group discussions FGDs were held. Three FGDs with parents of low social economic status while two FGDs with parents from high economic status. A written consent to participate in both Kiswahili and English languages, was distributed to the parents three weeks in advance to allow them to make a decision. During FGDs, there was a translator for Kikuyu language because majority of the participants from Kiambu country are from the Kikuyu ethnic community, which is the largest ethnic group in Kenya. The qualitative data from the focus group discussions were transcribed, coded and analyzed thematically. During the focus group discussion, tea and a snack were served as a compensation of the time for each participant.

### 4.4. Participant Observations

Observations documented as field notes were also analyzed to complement the study findings.

#### Research Ethics Clearance

A legally binding ethics code was signed between the researcher and the SARECO funding instrument (represented by University of Bern and Swiss Tropical and Public Health Institute). To obtain a host institution for the fellowship, the researcher sent a letter to the education department of the African Population and Health Research Center (APHRC). Dr. Ngware of APHRC, put the researcher in contact with the relevant representatives from Ministry of Education (MoE) in Nairobi about the request. An approval letter was obtained from MoE through Kenya Institute of Special Education, granting the right to conduct the study and authorizing the fellowship at KISE. With the authorization to conduct research, the researcher was able to draft letters of consent (in English and Kiswahili languages) which were distributed to parents of children with DDs in Kiambu and Nairobi counties. The target sample were parents that are already registered in the database at KISE.

## 5. Results

For this article, the researcher presents the results using an approach that focuses on the conscious experience of parents of children with a developmental disability, as the source of knowledge in relation to the changing situations and interconnectedness with their ability to socially integrate. The emphasis is on presenting the excerpts from the interviews and focus group discussions in contexts that facilitate the reader’s emphatic understanding of the subject. Interpretations are advanced when the material seems strong enough to warrant them.

### 5.1. Moral Model of Disability and Its Influence on Primary and Secondary Caregivers in Kenya

Social and Cultural aspects of community life have often shaped the way persons with disabilities are treated by others [29]. In Kenya, children and adults with DDs were often shunned by the community due to cultural beliefs [30]. In Kiswahili language the country’s second national language, a person who has a mental illness that cannot be diagnosed through medical assessments is known as “*mwenda wazimu*” or mad person. For the longest time, there were only two public healthcare facilities that would address mental healthcare [31], these are; Mathari Hospital in Nairobi and Port Reitz hospital in Mombasa. Perceptions about the causes of DDs may be attributed to witchcraft, depending on one’s ethnic background [32]. This prejudice is also noted for other African countries such as Tanzania [33], Malawi [34] and Namibia [35]. Prejudice results in stigma around persons with DDs and their caregivers.

In this study, caregivers are categorized as follows; primary caregivers (parents and close relatives such as aunts, grandparents) and secondary caregivers as the educational and rehabilitation practitioners of the children with DDs. Studies have shown that the primary caregivers of children with DDs in African countries such as Tanzania [36] may become immobilized by the around the clock care that they have to provide to their children [34]. This may negatively affect their own health [24]. During the focus group discussions parents revealed that *“the role of caregivers is increased according to the capabilities of the child. Children with severe developmental disabilities are often not brought for assessment because the parents have lost hope and some prefer to just hide the child in the house”* FGD, KISE low SES.

Another challenge lies in the absence of enough trained rehabilitation specialists. The research noted that there is only one speech therapist at KISE (a Catholic nun). She sees on average 10 children a day. Parents of children with autism come to KISE mainly to have access to the free speech therapy. In a focus group discussion, some parents explained that *“It is good to meet a therapist like the catholic nun because she is a devout Christian and during the session, we feel that she is also praying for healing for the child, even though she is not verbalizing it. This is very reassuring”* FGD, KISE, low SES.

Notably, the interaction of the catholic nun with the parents also shows the strength of the existence of the moral model of disability because parents feel that her religious background is beneficial to their childrens’ wellbeing.

The study observed that, the parents of children with autism, bring them for assessment to KISE at an average of 4–5 years of age. At KISE, the free speech therapy is only offered up to 6 years of age. However, because other DDs such as Down syndrome, are recognizable through physical features, parents of such children came for assessment earlier averaging between 2 years to 4 years. The choice to bring the child to KISE for assessments has not been made compulsory by the government and therefore remains the prerogative of the parent. In other discussions, parents revealed that *“At least when the child is capable of walking, or mumbling some words, we feel encouraged to bring the child here for assessment”*. FGD low SES, KISE, October 2018.

According to another group of parents “*Majority of those parents with Down syndrome who came to KISE wanted a placement for the child due to bullying at a pre-primary school. Whilst those with children who have autism wanted a warning letter for teachers who were frequently beating their children due to the perceived ‘rudeness’ of the child*” FGD low SES, KISE October 2018.

According to the FGD with parents from low SES, a child who cannot walk to school may automatically place him or her in the category of severe even though his/her cognitive abilities may be assessed as moderate. The respondents from low SES attribute this to the challenges faced by using public transport. The public buses and “*matatus*” (minivans for 12 passengers), do not have a ramp system to accommodate wheelchairs. In addition, respondents (mostly mothers) stated that they have to carry their children on the back to take them for rehabilitation or healthcare services or when visiting family members in other parts of the country. In addition, most schools do not have ramps for accessing classrooms or toilets. Therefore, those children with a DD who also use mobility aids such as wheelchairs are considered to be needier.

In addition, there are the cultural beliefs about the “cause” or source of the child’s disability that have serious implications on either the mother or father. These beliefs are not homogenous and are dependent on the parents’ ethnic community. For example, Gona et al. (2017) states that amongst the Giriama in Kenya, the mother is presumed to have been adulterous hence the presence of a developmental disability in the child. Whilst in some cases the blame is placed on the father. It is said that he had been adulterous and ‘touched’ the child after his transgression [32]. The disability that is believed to have been caused by a “witchcraft spell” from relatives or jealous neighbors is seen as bad luck, however it allows the parents to be absolved of blame. Overall, during this study, FGDs with parents established that when a child’s disability is believed to have been caused by external factors, such as the will of God or a jealous neighbor or relatives, it is measured as a lesser problem than that caused by beliefs accusing either the mother of father of being “adulterous” and therefore has lesser affiliate stigma [25], to the primary caregivers.

### 5.2. Background on Public Educational Institutions for Developmental Disabilities in Kenya

The need to review the Kenyan educational system has been viewed as integral to national development since the post-colonial government of 1963 [37]. In 2003, the Government of Kenya, following a 2002 election campaign pledge, introduced the Free Primary Education (FPE) policy in order to universalize access to primary education and increase educational attainment in the country [38]. This means that teachers in primary schools are overwhelmed by the enrollment numbers of ‘typical’ children and therefore are easily discriminatory towards those with a developmental disability. Even though the model of inclusive education is part of the sustainable development goals, in Kenya, there still exist schools which only cater for children with DDs. Some parents believe that this system protects their children from sexual abuses which has been noted in other African countries such as Botswana [39] and Malawi [40].

For insights on primary and secondary education, the researcher visited two government primary schools namely; St. Patrick’s school for the mentally handicapped in Thika town and the model inclusive pre-primary school. The latter is a perfect case study on how inclusion works, and is situated within the Kenya Institute of Special Education. The researcher also visited a privately run school called Mirema which has a special needs unit. Mirema was targeted because a mother whose son has autism, started a collaborative project on inclusive employment through a leading pharmaceutical company from the United Kingdom. For a look at tertiary education, the researcher visited Maria Magdalena sheltered workshops for persons with disabilities in Thika. This will be discussed in detail below.

#### 5.2.1. Kenya Institute of Special Education (KISE)

The Kenya Institute of special education was established on 14^th^ February 1986 to train teachers and other personnel who offer services to learners with special needs and disabilities. According to Chege (2016), children with disabilities who are from poor backgrounds in Kenya are less likely to have access to education [41]. Therefore KISE runs several educational assessment centers countrywide as well as a model inclusive pre-primary school in Nairobi, targeting Kenyans from all social economic backgrounds. In 2017/2018, there were 1577 children with special needs who were assessed. In addition, 2150 children with cerebral palsy were offered rehabilitation services at the institute’s assessment centres. As of 12 October 2018, there were 29,406 special needs education teachers and instructors who have graduated from the institution since 1986.

#### 5.2.2. Saint Patrick’s School for the Mentally Handicapped

The Saint Patrick’s School for the mentally handicapped in Thika is a government institution that receives support from the Lion’s club of Kilimambogo, a charitable organization. The school has a population of 67 boarders and 50 day-scholars between 5 and 25 years. The class hours run from 8.00 a.m. to 3.00 p.m. in the afternoon.

According to the head teacher; “*The school teaching staff is well equipped for children with developmental disabilities. What is lacking is funding to construct more classes because of the demand for enrollment from parents. The neighbouring special needs school –Joytown, which I used to be head teacher, has a capacity of 340 boarders. So I know that it is possible to manage that number. We have a waiting list of 250 children and this is very frustrating to parents. In addition, there are many special needs teachers now in Kenya and the teachers service commission pays them an additional 10,000 Kenya shillings on their fixed salary. This is a good incentive as compared to other teachers teaching in ‘typical’ schools. The will to teach children with DDs is there however, we need to get psychiatric staff, a resident nurse or even a clinic because these children are often sick. Can you imagine when a boarder is sick, we still have to rush him/her to a government hospital at night. There is no cost for treatment but we have to finance transportation, have an accompanying person in case of violent spasms from the sick child. Sometimes when the child does not have a NCPWD card, we queue like others with ‘normal children’ and this is frustrating for both the caregivers and the child with a disability*” KII, Head teacher.

The study established that, many parents opt to enroll their children in boarding facilities because they cannot drop-off their children at 8.00 am every morning and then report on time for work. Similarly, the pick-up time for children is at 3.00 pm and does not align to normal working hours. It is within this context, that the school offers boarding facilities. However, part of the caregiving role is transferred to the teachers and guardians of the boarders.

The increase in demand for facilities for persons with DDs has been brought about due to several factors. The *first* is; healthcare systems in Kenya are better than they were 20 years ago and therefore there is a higher life expectancy of children with developmental disabilities. A *second* factor is reliability of the family network; the rate of marriage breakdown has increased meaning that there is a shrinking pool of a family networks to help the primary caregiver (often the mothers in this case). *Lastly*, the notion of the ‘family’ especially in Nairobi and its environs such as Kiambu, has shifted from the extended family which was larger, to nuclear family which is smaller. Overall, this increases the caregiving role by immediate family namely; parents.

### 5.3. How the Moral Model of Disability Leads to Mistreatment of Children with Disabilities

Globally, there has and still continues to be maltreatment of children with disabilities [42]. According to Bunning et al. (2014), persons with disabilities often face abuse and mistreatment at the hands of minders, relatives and even in educational institutions in the area of Kilifi, Kenya [43]. The acts of violence in educational settings have also been reported in European countries such as Switzerland [44]. However, Switzerland is going a step further by creating a proper legal framework for what constitutes abuse and abusive treatments especially within the educational milieu [45]. The aim is to hold any perpetrator accountable.

This study shows that, according to Table 1, the majority of the violence committed to children with a DD, is perpetrated by other children who bully them at school. Also, the minders or house girls who are left to take care of the child at home, frequently beat the children. The children at primary school level are beaten more than their counterparts at secondary schools. Lastly, although low in number, there are children with DDs who also beat up other children.

According to focus group discussions; “*We (high SES) often have two working parents and we have to leave the child in the hands of the minder or house girl. The advantage of the private schools that we enroll them in, is that they have strict policies against bullying and beating of our children. Unfortunately at home, we have no control over the house girls, because without them we cannot go to our workplaces*” FGD High SES.

“*We are forced to employ ‘shadow’ teachers for our children. These are teachers for special needs education who have recently graduated from the institutions such as KISE and Kenyatta University. Most are young and do not have an official job posting yet, so, we provide them with accommodation and transport to and from the school where the child with special needs is enrolled. We also pay them a monthly salary of between 15,000 to 20,000 shillings a month. The shadow teacher accompanies the child to the ‘regular school’ so that the teachers there do not complain of being overworked*” FGD High SES.

According to the focus group discussions with parents; “*It is often the children with Down syndrome that are bullied because others think they are retarded. Our children are sometimes called “Kiogo or *kiũgũ*” meaning, mentally retarded in Kikuyu language. Also our children are mistreated at school because some of the teachers already feel overwhelmed by the large number of children enrolled in public schools, so when our children fail to obey the teachers instructions, they are thoroughly beaten or given punishment. That is why we come here to KISE to get a warning letter written to the school or sometimes we want a transfer letter for the child to a different school, so KISE writes a placement letter recommending a school for us.*” FGD low SES.

Another issue that was often discussed during the FGDs was the lack of educational certification for children with DDs. During an FGD, some parents whose children are enrolled in ‘regular’ schools stated that “*once our children reach class 8, the school requests the parent to withdraw the child from sitting the final exam. Some tell us that the child will not be registered to do the Kenya Certificate of Primary Education KCPE, because his/her score will lower the overall performance of the school. The public schools like to maintain a high KCPE score in order to attract new parents to the school. So our children are excluded so that parents of ‘typical’ children can find the school attractive and their enrollment rates can remain high.*” FGD High SES.

The government has gone a step further and started the model inclusive pre-primary school at KISE as a pilot institution for early childhood development. The research established that priority for placement is given to (i) parents who have more than one child with a DD and (ii) parents whose child with special needs has multiple disabilities.

According to an interview with a mother of two children with autism “*The school is convenient for me because I live in Kasarani and can bring them here every day. Tuition is also free and they can have free speech therapy with the Catholic nun very frequently.*” KII, 26 year old mother of two children with autism.

## 6. Medical Model of Disability

### 6.1. Medical Model of Disabilities and Its Implications on Developmental Disabilities

The medical model of disability which puts emphasis on disability as a “disease” means that historically, health practitioners in Kenya may have approached it as such. Interviews with medical specialists revealed that there needs to be a review of the way questions are framed by the healthcare institutions when conducting assessments with both primary and secondary care givers of children with developmental disabilities. Some of these specialists stated that the line of questioning may sometimes make the parent of the child feel guilty of having ‘committed a crime’ during the pregnancy or during the infancy stage of the child hence contributing to the disability. The line of questioning is embedded in the historical attitudes about disability in Kenya.

The researcher established that the preference for medical practitioners was based on two factors; *first*, many parents prefer to consult either male or female medical practitioners who are recommended by other parents of children with special needs. *Secondly*, sometimes the preference was based on the gender of the practitioner. The “preference” for female doctors by some parents (mainly mothers of children with special needs) also resonates with work about female patients’ preference for female doctors in Nigeria. A study by Saidu et al. (2015), has shown that there is a strong preference by female patients for consulting female doctors in North Western Nigeria. It also, emphasizes that the patients’ age, marital status and religion are very strong factors for the preference of female doctors [46]. An excerpt from an FGD with parents from high SES revealed the following; “*The Kenyan doctors at Mater Misericordiae Hospital have been performing heart surgeries on our children and sometimes invite doctors from India. There is a female doctor who is very popular amongst our group of parents. She is a very good doctor and she also gives us free counselling as mothers of the child*”.

The study established that there are specialized private facilities in Kenya such as Nairobi Hospital, MP Shah Hospital and Gertrude’s children’s home hospital, which offer support for ophthalmologic interventions, adenoidectomy, heart surgeries, orthopedic interventions as well as physiotherapy occupational and speech therapy. However, other parents prefer to take their children abroad specifically for heart surgery to hospitals such as Miot and Fortis in India.

A mother of a girl with Down syndrome describes the process as follows; “*I went to Miot Hospital in 2015, and the healthcare package for a heart operation was 10,000 US dollars. I paid from my pocket, with some help from crowdfunding by relatives. However, just this year (2018), the National Health Insurance Fund, NHIF is now paying for half the cost. This is very good for my fellow mothers whose children require heart surgery. I preferred Miot Hospital because at Miot, they give you (the caregivers) a place to sleep inside the hospital. Actually, you do not have to look for a hotel outside like in some other hospitals. This reduces costs as well as distance for accessing your sick child.*” KII, High SES, Mother of girl with Down syndrome.

In this study, 70 respondents did not take their children abroad for health interventions and preferred to go to local hospitals. However, 2 respondents took their children to Dubai, 3 to the USA and another 2 to Europe. Amongst the hospitals and cases highlighted were; one child with cerebral palsy was taken to Evalina Children’s Hospital in the UK and another with Down syndrome was taken to Universitätsklinikum in Erlangen, Germany. The links to obtain the heart surgery were obtained through family networks and church organizations. 13 other respondents took their children specifically to India for heart surgery. This is because the government of Kenya and India have good working relations in terms of Healthcare. Since 2018, the NHIF covers heart surgeries performed at some hospitals in India such as Miot. Sometimes doctors from India are invited to Kenya to work with cardiologists in hospitals such as Mater Misericordiae and Kenyatta.

For parents from low social economic status, the only affordable healthcare is offered at Government hospitals such as Mathari and Kenyatta. Free assessment and rehabilitation services for developmental disabilities are offered at the Kenya Institute of Special Education. The study noted that it is the rehabilitation services such as speech therapy, which pose a major financial setback to parents because these are not covered under the typical healthcare packages. Due to the challenges brought on by a myriad of factors; social stigma, cultural stigma, financial role, and sometimes exclusion from education facilities, many parents may be forced to abandon children with DDs at government run childrens homes. For example, the Rescue Dada children’s home in Ngara (Nairobi) also has several children with developmental disabilities who are brought to KISE for assessment and rehabilitation services. Abandoned children are also brought to private children’s homes and orphanages. The researcher visited the Kamili children home and NEST situated in Nairobi.

### 6.2. Disability Assessments; the Role of Government Hospitals such as Mathari

Surveys have been done to establish the challenges that persons with disabilities in middle income to poor countries face when trying to access healthcare [47,48]. For this reason some countries such as Kenya urge parents of children with disabilities to do disability assessments at Government hospitals so that they receive a card issued by the national council for persons with disabilities (NCPWD). By presenting this card; parents can be fast-tracked at public facilities i.e., exempted from long queues, exempted from paying tax on educational materials when arriving from another country, exempted from tax on imported nutritional supplements, eligible for government tuition fee scholarships, eligible for government cash transfers for severe disabilities and tax exemptions if they own or run an income generating activity for their child with a disability.

One of the designated hospitals for these disability assessments is the Mathari Hospital which was for the longest time the only government facility that offered psychiatric help to Kenyans. Since 2014, it was upgraded to be a national referral hospital. It also runs the largest drug rehabilitation centre in Kenya and has highly skilled and well trained medical practitioners. The services are very affordable; for example; consultation fee for a child costs 50 Kenya shillings (0.50 Swiss centimes). A disability assessment costs 850 shillings about (approximately 8 Swiss francs).

According to focus group discussions with parents at KISE “*Mathari is affordable. Before the medical consultation with the doctor, the hospital organizes a free consultation session with a resident psychologist or counsellor. This helps us to share our fears and uncertainties about the situation. However, I only wish there could be a children’s hospital catering for developmental disabilities and for mental healthcare that is separated from the adult facility. This is because when walking inside Mathari, I saw an adult patient with schizophrenia fighting with a guard and my son was scared.*” Response during an FGD with parents at KISE.

“*Yes, that is true, for me I saw an adult male trying to climb onto the barbed wire that is fixed on the perimeter wall that protects the adult male ward in Mathari, and this adult was completely naked. I had to cover the eyes of my child as we were passing through*”. Response during FGD with parents at KISE.

There are several elements here; first, the caregivers’ challenge for DDs lies in the cost of rehabilitation and healthcare as illustrated through the focus group discussions and interviews. The cost of medical interventions such as heart surgeries is being catered for by NHIF. However, rehabilitation services need to be included in these costs as well. Second, there needs to be a construction of a separate medical facility for children with DDs far away from the vicinity of the boarding facility for adults at the Mathari mental and referral hospital. This will be less traumatic for both children with DDs and their parents and may encourage more parents to take children for assessments. The positive interpretation of the situation is that the government has acknowledged that the issue of rights for persons with disabilities is important to the economic development of the country and as such has made it mandatory for disability assessments to be conducted at public hospitals. It has set up the national council for persons with disabilities council (NCPWD) to issue disability cards once these assessments are done and this will be discussed in the following section. 

## 7. Human Rights Model of Disability: National Council for Persons with Disabilities (NCPWD)

The Disability Act was passed by parliament in December 2003 and revised in 2012. In 2004, the NCPWD established. The NCPWD is responsible for conducting research on various disabilities issues in the country. At the moment, there is need for more statistical data about the impact of disabilities on the lives of children, for appropriate government planning and allocation of resources and for improvement of services for persons with disabilities.

The NCPWD is generally working towards a barrier free and disability friendly environment. The Kenyan constitution that came into effect in 2010, established a new model of state government. The country which until then had provincial administrative boundaries known as provinces, is now structured under county governance system. This new devolved system of governance came into force after the general election of March 2013 [49]. The NCPWD works at county level through the county social development officers. It has also taken up the task of registration process of persons with disabilities and organization representing the needs of persons with disabilities.

According to Table 2, most respondents with a child with DDs had not heard of the card until when they came to KISE for assessment. However, as aforementioned, there are many advantages for the NCPWD card holder and his or her caregivers.

The Table 3 shows that a substantial number of respondents (42) wanted reforms to introduce a school curriculum (at primary and secondary level) for evaluating children with DDs. These were mainly respondents whose children had reached the level to sit for the Kenya certificate for primary education (KCPE) and yet their children were not enrolled for this national exam. There was also a preference by 21 respondents for the introduction of special needs units in schools. 32 respondents are satisfied with existing government health reforms. It is for this reason that fewer respondents wanted additional health reforms such as; medical subsidies for treatment of illness associated with development disabilities among children, reforms on mental health policy and removal of visa restrictions for medical treatment abroad as shown in Table 3.

### 7.1. Right to Inclusive Education—Mirema School

In Kenya, enrolling children with special needs in private institutions has high cost implications. The study established that at Mirema School, there is a special needs unit called ‘differently-abled’ unit. To enroll a child, one has to pay an assessment fee of 5000 Kenya shillings, registration of 6000 Kenya shillings, tuition of 30000 Kenya shillings, tuition pre-vocational of 40,000 Kenya shillings, lunch per term is 6000 shillings, swimming activities at 6450 Kenya shillings, mentorship training at 1000 Kenya shillings, and transport to and from home depending on your residential area is between 6000 to 1000 Kenya shillings per school term. Therefore, the school fees is approximately between 45,650 to 54,850 Kenya shillings a term depending on whether the child is under the vocational training. This is approximately 500–600 Swiss francs, a cost which is high for the average Kenyan. Nevertheless, there are parents who prefer to enroll their children with special needs at private schools. Some parents go a step further and create employment opportunities between the schools and the private sector.

This is illustrated below by a Kenyan mother who began voluntary work at Mirema School as a way to cope with the DD of her son. It has been noted by Trute et al. (2010), that mostly, it is mothers who bear the emotional role of care for children with special needs and most of them look for coping strategies [50]. An interview with the mother revealed the following; “*When I visited the school, the special needs unit were teaching the children bead work and other crafts. I thought that there was need to diversify their capabilities So, I created a job for these fifteen young adults with autism, all were above 18 years but were still enrolled at Mirema. They were employed at the pharmaceutical’s manufacturing site on produce packing line or rather, to pack tablets into the blister packs. They were on a daytime four hour shift. I initiated this project because of my personal experience of having a son with Autism. I wanted to engage in corporate social responsibility by drawing from my own experiences. I felt inspired to start this project because of my son. I needed to create an avenue where corporate Africa can embrace inclusive employment*” KII, Mother of a boy with autism, who is also a CEO for an international pharmaceutical company in Kenya.

### 7.2. The Right to Gainful Employment—Maria Magdalena Sheltered Workshops

There have been debates about the need to include persons with disabilities into forms of gainful employment [51]. At the Maria Magdalena sheltered workshops for persons with disabilities, the research study established that there are 39 boarding trainees and 4 ‘dayscholars’. This is an increase from 2015 where there were 32 in total. These trainees do metalwork, woodwork and agricultural work. The trainees sell wooden school desks to local schools in Thika area and are paid a labour fee. The trainees also run the local cafeteria which is called “*Lindenhof*” named in recognition of the efforts by a German catholic priest who started the institution. The annual tuition fees is 28,000 Kenya shillings (280 Swiss francs), which is inclusive of boarding facilities. The institution has a long waiting list because there is no official cut-off age, the oldest trainee at the time of research was 32.

According to the trainers at the sheltered workshop “*we are very strong in metal work and carpentry. However there needs to be a training programme on computer use initiated. We are looking for funds for this.*” KII with a Trainer at Maria Magdalena.

As acknowledged by Mizunoya and Mitra (2012), there is a gap in employment rates for persons with disabilities [52]. This sheltered workshop plays an important part in emancipating the young adults. However, there is scarcity of such facilities in Kenya (centres with boarding facilities are Sikwiri in Kisii and another in Nyahururu).

Another gap is in the area of sports inclusion. It has been noted by Nagel et al. (2018), that sports is one avenue to promote the social integration of marginalized groups in the society [19]. The research established that the institution is not involved in competitive sports for persons with disabilities such as the Paralympic games even though the trainees expressed interest. According to a Selina a 24 year old lady with autism who is working at the institution “*it can be nice if we also compete in national and international competitions for running, so we can win some trophies and appear on TV*”.

The trainers attribute this to the absence of funding for sports organizations for persons with DDs. “*Some organizations do not give our people with developmental disabilities a chance at competing in sports at national and international level. There are more opportunities for persons with physical disabilities. All you see on television are those athletes with visual impairment like Henry Wanyoike or Henry Kirwa and the javelin/shot put winner- Mary Nakhumicha Zakayo, but never someone with a developmental disability*” KII, Trainer, Maria Magdalena sheltered workshops. The trainees expressed interest in competitive sport at Paralympic level, specifically, inclusion in athletics, for which Kenya is globally renowned for.

## 8. Why Would Parents Chose to Migrate?

### 8.1. Urban to Rural Migration

The 2018 KISE survey found that 72.6% of children with disabilities, live in the rural areas while 27.4% live in urban areas [12]. These statistics are important for this study because they indicate that disability is more prevalent in the rural areas, a trend that also coincides with the findings of the 2008 Kenya national survey for persons with disabilities. However, the statistics may also indicate another hidden issue; some parents in urban areas may prefer to migrate back to the rural areas once they have a child with a disability. During focus group discussions with parents of children with DDs, it was established that some of them would migrate back to rural areas so as to receive social support from close relatives such as grandparents in raising the children.

In addition, some parents especially those with adult sons, noted that there are more job opportunities for young men in the rural areas. “*In the countryside, our boys can find jobs such as herding cattle, cleaning poultry pens or taking care of pigs. Some are even asked to cut napier grass and to feed the cattle*.” FGD low SES parents at KISE.

Those parents with daughters had a different reason as to why they would migrate from the urban areas to the rural areas “*In the rural areas, everyone knows everyone, so if one of our children (with a developmental disability) is impregnated, the culprit will be known and will face social justice. In the rural areas, people monitor each other’s movements. It is not like here in Nairobi, where I can live in a flat (apartment) for many years and not even speak to my neighbor*”. FGD low SES parents at KISE.

Another reason respondents gave as incentive to migrate from urban to rural areas is because of the cost of educational facilities. In rural areas, boarding facilities (public and private) for children and young adults with special needs cost far less than those in urban towns such as Nairobi. A survey conducted in 2014 by the *VSO Jitolee* [53], a non-governmental organization, established that there are more children with disabilities (CWDs) out of school than those without disabilities. The survey also found home-based and systemic factors that hindered CWDs’ school attendance, persistence of stereotypes, misconceptions, stigma and discrimination towards children with disabilities in the schools and community.

These surveys are important for this study because, for several years, developed countries have been working steadily towards a policy of inclusive education [54], and Kenya has recently started to catch on. The concept of inclusive education presupposes that the education of learners with disabilities should be in mainstream schools where they and other children learn together. However, a study by Ong’era (2011), has shown that children who exhibit emotional and behavioural difficulties (EBD) in Kenya present a challenge for parents, teachers and significant others [55].

The Kenyan government encourages educational institutions to embrace inclusive education through a new policy framework [56]. In contrast, Switzerland still maintains some centres for special needs education in some cantons such as Jura and Bern. In such centres, learners with DDs are segregated from ‘typical’ children. However, emphasis is being placed on the importance of such ‘specialised’ centres for the protection of children with DDs against abusive practices that would otherwise happen in a ‘regular’ school [44].

### 8.2. Global Migration

This study has implications on global policies related to disability and migration because of four reasons. *First*, it has been observed that amongst Somali immigrants in Scandinavian countries such as Sweden [57], there is a high prevalence of developmental disabilities such as Autism in children. According to Baker and Kim (2018), several theories have been put forth to explain this phenomenon in the United States of America, including: vitamin D deficiencies caused by the relocation from an equatorial region to northern climates with scarce sunlight; consanguineous marriages, and duplicate vaccinations due to time spent in refugee camps and transnational migration ([58], p. 2).

*Secondly*, there are Kenyan born-Somalis who form an integral part of the population. According to the Kenya Integrated Household Budget Survey published in 2018, Somali families in Kenya are at least one and a half times bigger than the average household in Kenya, and twice as much as the families in Nyeri, Nairobi, Mombasa and Kiambu counties. Wajir, Mandera and Garissa counties, which are home to most Kenyan Somalis, have between six and seven children on average.

Wajir county has the biggest Somali families at 6.6 children per household, followed by Mandera 6.4 and Garissa 5.5 members per household [59]. In the last census, population growth in North Eastern Kenya (where majority of Kenyan Somalis reside) rose almost three-fold, from 962,143 in 1999 to 2.3 million in 2009 [60]. It is important to create awareness amongst the Somali community who like other Kenyan ethnic groups, believe that mental health illness or disability is caused by “waddado” or spirit possession [61].

*Third*, Kenya is known to be a conduit for refugees from neighbouring countries such as Somalia, Sudan and Ethiopia who may be seeking to settle in Europe. It has been established that the traumatic experiences that refugees undergo during migration may lead to them to develop psychosocial issues in the new country of destination. A study by Chatzidiakou et al. (2016) in an emergency department of a Swiss hospital shows that these issues are mainly acute psychiatric health problems. In women, the most common reasons for admission were social problems, followed by depression and psychosis. In men, psychosis was the most common reason for admission, followed by auto-aggression and drug abuse [21].

*Lastly*, a study amongst migrants in Sweden has linked mental illness such as depression as a contributing factor to Autism in utero (during pregnancy) [62]. Kenya hosts the largest refugee camp in Africa called ‘Daadab’ and it borders Somalia. It is also in close proximity to the Ethiopian border. In Daadab, several nationalities including but not limited to Ethiopians and Somalis have been identified. Findings from this study can help the Kenyan government to assist Kenyan ethnic communities as well as the refugee communities that may be facing challenges related to disability. Overall, challenges associated with caregiving, may motivate parents to think of migrating to developed countries as indicated in the table below.

According to Table 4, a total of 66 out of 90 respondents (combination of 4 categories; chain migration economic migration through a corporate or multi-national job, migration through domestic work and migrating to abscond parenting) are interested in migrating without the special needs child. 62 of them would send remittances for the up keep of the child. On the other hand, 24 respondents (combination of 3 categories; setting up an international business, higher education and international jobs) would migrate with the child. In some categories, such as chain migration, there were many respondents from the low SES who are willing to migrate to work as a ‘house help’ for a wealthy relative living abroad. This was a surprising element in the research.

According to focus group discussion “*The ideal migration pathway is to join a relative abroad because that way there is already a social network to support the parent in the new country.*” FGD High SES.

According to an interview with a parent of a young adult with Down syndrome “*I migrated to the USA when my daughter was starting school. I deliberately chose a position with the World Bank there because I knew that my daughter would receive a good education. My wife and I just wanted to stay there until she was old enough for us to look for other options on our return to Kenya. Our goal was never to settle permanently in the USA. That experience was very fruitful because her former school in US still invites her for conventions and she is a good public speaker. She even receives awards for this. She uses that platform to motivate her fellow young adults back home in Kenya*” KII, Parent of daughter with Down syndrome.

This study indicates that parents from low social economic status were interested to migrate for a temporary period where the parent can send remittances to the secondary caregiver of his/her child. This is to facilitate access to private healthcare and private rehabilitation services for the child with a developmental disability. Another reason to migrate was to abscond parental duties. A mother from a low SES revealed during an interview that her husband took up a job as an acrobat in the United States of America just to abscond the responsibility of their child born with cerebral palsy. Although there were only four such cases, it shows the effect of DDs on parenting.

Another interview with a guardian (Aunt) of a boy with autism and who is from a low social economic status revealed the following “*I went to Dubai to work as a house help. However, when I reached there, my kafeel or sponsor* [63] *placed me to take care of children in an institution for children with autism at Sharjah. At first I was scared, then after working there for a while and receiving a very good salary I was grateful for the opportunity. I learned a lot and was paid very well. When I came back to Kenya, I was the only one in my family who can get along with my sister’s son who has autism. I am happy about that experience in Dubai because it taught me a lot especially patience*.” Guardian (Aunt) of a boy with autism, low SES. 

These revelations from both the World Bank employee who is a high SES respondent and the former employee of a special needs children’s home in Dubai who is a low SES respondent, show that there can be both economic and educational benefits derived from temporary migration.

## 9. Discussions and Conclusions

This study draws attention to how the social environment in Kenya is challenging to the caregiving role of parents who have a child with a developmental disability. In this article, the researcher takes a step forward and looks at the caregiving theory [22] and theory of social integration [23] to understand what it means to deal with disability in Kenya. Different models of disability; medical, moral and social [27], are introduced to show the disjuncture between policy and practice. Discrimination of persons with DDs may happen because they are considered incapable of using their intellect in a manner that is seen as productive or in accordance to societal expectations.

In Kenya, persons with DDs may sometimes directly or indirectly lack the following opportunities; remunerative jobs, the right to public transport, right to affordable healthcare, right to education or even the right to marry. Due to the social and cultural beliefs found in many ethnic communities, disability in Kenya is treated as a taboo subject and therefore parents have often been marginalized. The social, cultural and economic context may present a different set of challenges for a parent or caregiver residing in Kenya as compared to one in a developed country such as Switzerland. Most respondents in the study agreed that a special curriculum should be developed for children with DDs. This will ensure that persons with DDs obtain a certificate at primary, secondary and tertiary levels of education. This in turn may increase their chances at gainful employment.

The human rights model of disability concentrates on reform of laws rather than social transformation. It is imperative that the approach towards handling issues on DDs in Kenya adopts some of the principles behind the social model of disability in order to create an enabling environment for children with DDs. This article shows that the social and financial aspects of caregiving for children with DDs rests wholly on the parents. Even though these parents may belong to different class systems, whereby families from a high social economic status can afford education and healthcare for their children, they (high SES) have to incur additional costs just to ensure that their children are treated like ‘typical’ children. For example, having to employ ‘shadow teachers’ so that their children can have access to a school with ‘typical’ children. Furthermore, the secondary caregivers in educational institutions such as St. Patrick’s school for mentally handicapped in Thika, are faced with challenges when their boarders fall sick. Therefore, it is the prerogative of the parents to apply for an identity card from the NCPWD so as to enable faster access to healthcare and rehabilitation services in government institutions.

The study demonstrates that the interest to migrate by caregivers of children with DDs, is driven by uncertainty about the healthcare, education options and social exclusion of their children. The choices and reasons given to migrate are constantly changing. The interest to migrate ranges from; domestic (from urban to rural areas) to international (from a less developed country to a developed country). This study acknowledges findings from other studies demonstrating that communities which practice co-sanguinity such as Somali, are at high risk of having children with DDs [57,58]. Traumatic experiences by refugees may lead to babies developing autism in utero [62]. KISE has to increase outreach services to the Kenyan Somali community. In addition, KISE may have to extend its mandate and offer services to Somali refugees at the Daadab refugee camp in the north.

The study established that, as much as there are structures in place that can aid persons with disabilities, the practitioners are overwhelmed by the demand for their facilities and there is need for expansion. However, a more positive interpretation shows that, ever since the enactment of a new constitution in 2010, Disabilities Act 2003 and establishment of the NCPWD in 2004, an enabling legal environment for persons with disabilities exists in Kenya. It is this legal environment that may lead to social and cultural transformation, thus curtailing migration.

### Study Limitations

The findings are not to be used to generalize the situation of developmental disability in Nairobi and Kiambu as homogenous rather, some aspects were singled out as being specific to different categories of parents or guardian depending on (i) their social economic status and (ii) the severity of the developmental disability of the child.

## Figures and Tables

**Figure 1 ijerph-16-01010-f001:**
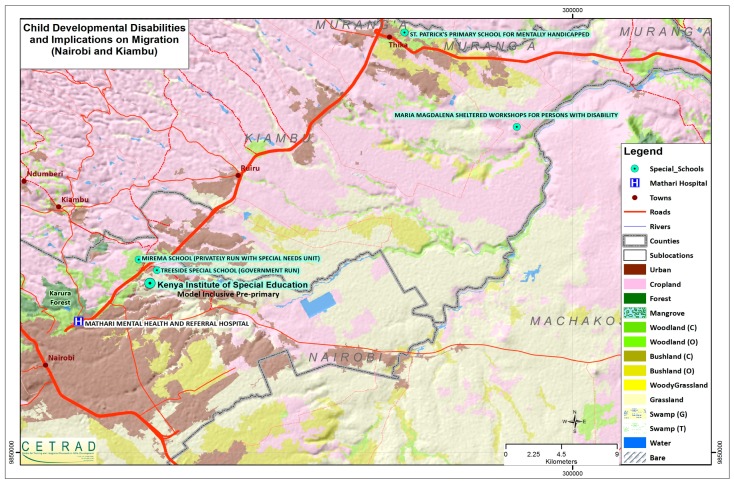
Map of area showing key institutions that deal with developmental disabilities.

**Table 1 ijerph-16-01010-t001:** Acts of violence committed against or by special needs children.

Acts of Violence Committed Against or By Special Needs Child	Observations	Rank Sum
Special needs child frequently beats up other children at school	7	311.50
Frequent beating by house girl (minder/maid) at home	16	757.00
Frequent beating by parent at home	4	178.00
Frequent beating by teacher at primary school	12	534.00
Frequent beating by teacher at secondary school	1	44.50
Frequent bullying by children	24	1068.00
Frequent punishment at a primary school	16	757.00
Frequent punishment at a secondary school	10	445.00

Chi-squared with ties = 3.666 with 7 degrees of freedom. Probability = 0.8173. So, the *p* value 0.8173 is greater than 0.05. Observations: children with special needs are bullied by other children even though a few of them also beat up other children.

**Table 2 ijerph-16-01010-t002:** Knowledge about the NCPWD ID card by parents of children with DDs.

Knowledge about the NCPWD Identification Card	Observations	Rank Sum
Never heard of the ID card	60	2715.00
Child has the ID card but unemployable, had to open an IGA for him/her	24	1113.00
Have heard about the card, but not applied for card from NCPWD	3	133.50
Child has ID card and can access education at reduced cost (public school)	2	89.00
Child has ID card and can access healthcare at reduced cost (public hospital)	1	44.50

Chi-squared with ties = 0.632 with 4 degrees of freedom. Probability = 0.9594. So, the *p* value 0.9594 is greater than 0.05. NCPWD: national council of persons with disabilities; IGA: income generating activities, include beadwork or hairdressing for young women and carpentry work for young men with DDs.

**Table 3 ijerph-16-01010-t003:** Recommended reforms by parents.

Recommended Reforms by Parents	Observations	Rank Sum
Adding special needs units in primary, secondary and tertiary school systems	21	979.50
Increase national coverage/televised programmes on developmental disability	6	267.00
Introduce a school curriculum (at primary and secondary level) for evaluating intellectual disability	42	1869.00
Medical subsidies for treatment of mental health illness among children	8	356.00
Outreach and medical camps related to special needs done in rural areas	4	223.00
Recognize the need for a national association of persons with intellectual disability	2	89.00
Reforms on national policies on mental health in Kenya	2	89.00
Remove visa restriction for medical treatment abroad	5	222.50

Chi-squared with ties = 11.522 with 7 degrees of freedom. Probability = 0.1174. So, *p* value 0.1174 is greater than 0.05.

**Table 4 ijerph-16-01010-t004:** Potential Migration pathways of parents.

Potential Migration Pathways	Observations	Rank Sum
Chain migration through family networks (to send remittances for special needs child)	21	934.50
Migration through corporate or multinational job (to send remittances for helping special needs child)	20	890.00
Economic migration - seek domestic servant or construction work in Dubai (to send remittances to support special needs child)	21	979.50
Personally setting up an international business that facilitates migration for special needs child	6	312.00
Migration through admission into an institution of higher education (to facilitate migration for special needs child)	6	267.00
Migration through international jobs (to migrate with special needs child)	12	534.00
Migrating to abscond parental role of caring for special needs child	4	178.00

Chi-squared with ties = 7.730 with 6 degrees of freedom. Probability = 0.2586. So, the *p* value 0.2586 is greater than 0.05.

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
