# Peer review of "Child Developmental Disabilities, Caregivers’ Role in Kenya and Its Implications on Global Migration"

_ijerph, 2019, doi:10.3390/ijerph16061010_

Round 1

Reviewer 1 Report

This paper attempts to explore the experiences of caregivers and children with disability in Kiambu County, Kenya in relation to global migration. Overall, the paper needs revision to present a coherent and cohesive story about the research.

General comments

Please use people first (autism is an expectation and many people on the autism spectrum prefer autistic people) and neutral language when talking about disability. Avoid emotive terms like ‘suffers’. It is not necessary to capitalise Intellectual disability etc. (expect for Down syndrome)

Have a think about how you are framing disability and caregiver burden. It reads as if the children with disability are burdensome on families – I think you are trying to convey a more nuanced understanding – the environment is burdening parents which is why families are migrating. It may be useful to acknowledge (via a footnote potentially) about the limitations and negative connotations around the word ‘burden’. A good rule of thumb around your use of language is to think ‘if I was this person, would I like to be described this way?’ – most of us would not like to be described as a burden, but would prefer to acknowledge we need additional supports not available in the current environment, which causes ‘burden’ to myself/the people who support me.

Please be consistent in your terminology – there is constant switching between describing ‘parents or care-givers’, ‘parents’ and ‘care-givers’ – it might be useful to state the first time you talk about parents/caregivers that you will use the term caregivers to describe xxx due to xxx (I imagine you are also trying to capture others who support children with disability – grandparents, when children are relinquished, etc.). Also, be consistent in your use of a hyphen in caregivers.

You need to comment on the language you use – mental disabilities – and what this actually means in the intro.

The language and writing style needs to be revised – some writing is informal (e.g., ‘so called mental disabilities, line 30)

I recommend following the journals preferred layout for an article – background, methods, results and discussion. Each of these sections should be clearly defined. Currently, the methods morph into the results without any clear separation (as a reader this is confusing). There is content woven throughout the article which would be better placed in the background.

Please revisit the referencing style of the journal. You have used a mix of author-date and numbering style

Please revisit your use of footnotes. Most of the footnotes would be better incorporated into the text or are references. An example of a good use of a footnote is footnote #5

Title

The current title does not match the content of the article – once you’ve reviewed the article come back and check the title still fits

The footnote for the first author is not necessary. I was excited to read the methods and find out how your team managed this unique perspective but it was not mentioned at all. If this is key information and it impacted the study as you indicated, please discuss in the methods.

Abstract

Please revise. It may be useful to look at other articles in this journal to see how the type of things that need to be discussed in an abstract. Currently, it is full of unnecessary information for an abstract. There is also information in the abstract that is not mentioned again in the article – an abstract needs to summarise the article and make the reader want to look at the full text. It should not present information that is not discussed again in the full text of the article.

Some examples of what needs to be changed:

There are some definitions here (e.g. the first sentence about disability). This needs to be moved to the background section. Also, please revise your definitions – I think you should use the term ‘developmental disabilities’ as this is an umbrella term for all of the conditions you list as either an intellectual disability or learning disorder (please note, cerebral palsy is not an intellectual disability. You also focus on Down syndrome, but you can have an intellectual disability without having Down syndrome – did you specifically focus on Down syndrome, if so, need to discuss this). Indicate to the reader that developmental disabilities are commonly referred to as mental disabilities in Kenya. Because of the language difference, you may also need to explain (perhaps a footnote) if this also includes mental health diagnoses (e.g., depression, anxiety, schizophrenia, etc.)

I was interested to read about the conceptual model of caregiver’s burden, but this was not introduced or explained in the full text of the article and you have not referenced any literature for the reader to find out more about this model

Missing from the abstract is ‘why is this research important’

Keywords

The keywords do not match the title or abstract. The article is about a broader population than intellectual disability – include developmental disability. It is unclear the importance of education from this first page.

Background

Need to define mental disabilities (see above comment)

The background does not set the reader up for the rest of the article. It needs to explore the key things the research covers (disability, migration, education) + give contextual information about Kenya and disability in Kenya and Africa broadly. This should be clearly signposted and each section linked to the overall research question.

Make sure to keep this section to background/contextual information – remove reference to what the study found (e.g., page 4, lines 150-152, the study noted…)

The theoretical framework section needs to be strengthened, it is a mix of background information and the framework/approach you used. You should also incorporate some discussion about the framework/approach in the discussion section.

Methods

As a general guide, you need to provide enough detail in the methods so that someone could replicate the study. Currently, it is not clear how you conducted the research

As this research was mixed methods, I anticipated a discussion about how the different methods informed one another/how the different methods were brought together to create a cohesive story about children with mental disabilities in Kenya

Need to step through the different methods you used clearly. There were 3 different approaches (survey, interviews, and focus group) but then I reached section 6 and it was unclear what the purpose of this section was, was this information collected from the research and how it fitted with the research.

I find the research questions overly complicated, can it be written more clearly and concisely? This is also the first time I think health is explicitly mentioned (I may be wrong, but up until now, I hadn’t realised health was part of the research question)

I would suggest renaming the sub sections of the Methods to something more meaningful (e.g., 5.1 Survey of parents, 5.2 Key informant interviews, 5.3 Parent focus groups)

You need to include a discussion of the data analysis. Key information missing is how you analysed the qualitative data and how you managed/analysed the different data you collected.

Sections 5.3.1-5.3.4: I’m not sure what the purpose of these sections are? It starts off as describing the participants but then draws on literature and then starts presenting qualitative findings. Need to make it clear to the reader what is happening here

Need to include information about ethical approval and consent

One the first page, the first author has a footnote about being a parent and bringing two perspectives to the research, this needs to be discussed in the methods

Section 6/Results?

From the first few sentences of this section, this looks like a case study about two schools and a sheltered workshop, but how did the researchers come to write these case studies? Was it part of the data collection process? Part of the analytical process? Was this the aim of the study to describe the different settings children with mental disabilities find themselves in which would inform the research question?

I have similar concerns about section 7 and section 8. How do these sections fit in the research? If the purpose was to create case studies, this needs to be discussed in the methods and then signposted in the results when the case studies begin.

Discussion

Generally, the discussion should summarise the key findings, discuss how these relate to the literature, limitation, future research/policy recommendations. The discussion in this manuscript would benefit from a clearer structure like this

Make sure to clearly state what this article has added in the discussion and what should the next steps be?

Author Response

RESPONSE TO REVIEWERS

General comments

1.      Please use people first (autism is an expectation and many people on the autism spectrum prefer autistic people) and neutral language when talking about disability. Avoid emotive terms like ‘suffers’. It is not necessary to capitalise Intellectual disability etc. (expect for Down syndrome) – Rectified this

2.      Have a think about how you are framing disability and caregiver burden. It reads as if the children with disability are burdensome on families – I think you are trying to convey a more nuanced understanding – the environment is burdening parents which is why families are migrating. It may be useful to acknowledge (via a footnote potentially) about the limitations and negative connotations around the word ‘burden’. A good rule of thumb around your use of language is to think ‘if I was this person, would I like to be described this way?’ – most of us would not like to be described as a burden, but would prefer to acknowledge we need additional supports not available in the current environment, which causes ‘burden’ to myself/the people who support me –Removed the care-giver burden model

3.      Please be consistent in your terminology – there is constant switching between describing ‘parents or care-givers’, ‘parents’ and ‘care-givers’ – it might be useful to state the first time you talk about parents/caregivers that you will use the term caregivers to describe xxx due to xxx (I imagine you are also trying to capture others who support children with disability – grandparents, when children are relinquished, etc.). Also, be consistent in your use of a hyphen in caregivers.- Rectified

4.      You need to comment on the language you use – mental disabilities – and what this actually means in the intro.- Using the developmental disabilities language only.

5.      The language and writing style needs to be revised – some writing is informal (e.g., ‘so called mental disabilities, line 30)  Removed informal language

6.      I recommend following the journals preferred layout for an article – background, methods, results and discussion. Each of these sections should be clearly defined. Currently, the methods morph into the results without any clear separation (as a reader this is confusing). There is content woven throughout the article which would be better placed in the background.- Have adapted to journal style.

7.      Please revisit the referencing style of the journal. You have used a mix of author-date and numbering style – Done

8.      Please revisit your use of footnotes. Most of the footnotes would be better incorporated into the text or are references. An example of a good use of a footnote is footnote #5 – Done.

Title

The current title does not match the content of the article – once you’ve reviewed the article come back and check the title still fits- Revised title

The footnote for the first author is not necessary. I was excited to read the methods and find out how your team managed this unique perspective but it was not mentioned at all. If this is key information and it impacted the study as you indicated, please discuss in the methods.- Removed footnote.

Abstract

Please revise. It may be useful to look at other articles in this journal to see how the type of things that need to be discussed in an abstract. Currently, it is full of unnecessary information for an abstract. There is also information in the abstract that is not mentioned again in the article – an abstract needs to summarise the article and make the reader want to look at the full text. It should not present information that is not discussed again in the full text of the article. –Shortened the abstract.

Some examples of what needs to be changed:

There are some definitions here (e.g. the first sentence about disability). This needs to be moved to the background section. Also, please revise your definitions – I think you should use the term ‘developmental disabilities’ as this is an umbrella term for all of the conditions you list as either an intellectual disability or learning disorder (please note, cerebral palsy is not an intellectual disability. You also focus on Down syndrome, but you can have an intellectual disability without having Down syndrome – did you specifically focus on Down syndrome, if so, need to discuss this). Indicate to the reader that developmental disabilities are commonly referred to as mental disabilities in Kenya. Because of the language difference, you may also need to explain (perhaps a footnote) if this also includes mental health diagnoses (e.g., depression, anxiety, schizophrenia, etc.)- Throughout the article I use ‘developmental disabilities’. Have also made reference to the policy on Mental Health in Kenya to differentiate disability from mental health.

I was interested to read about the conceptual model of caregiver’s burden, but this was not introduced or explained in the full text of the article and you have not referenced any literature for the reader to find out more about this model- Removed the emphasis on caregivers burden because of the negativity associated with the word burden.

Missing from the abstract is ‘why is this research important’ – Have rectified this.

Keywords

The keywords do not match the title or abstract. The article is about a broader population than intellectual disability – include developmental disability. It is unclear the importance of education from this first page. – Have included developmental disabilities.

 Background

Need to define mental disabilities (see above comment) -  removed mental disabilities from article.

The background does not set the reader up for the rest of the article. It needs to explore the key things the research covers (disability, migration, education) + give contextual information about Kenya and disability in Kenya and Africa broadly. This should be clearly signposted and each section linked to the overall research question. – has been clarified.I have categorized the research into three parts (social model of disability, medical model and human rights model.)

Make sure to keep this section to background/contextual information – remove reference to what the study found (e.g., page 4, lines 150-152, the study noted…) - Done

The theoretical framework section needs to be strengthened, it is a mix of background information and the framework/approach you used. You should also incorporate some discussion about the framework/approach in the discussion section.- Strengthened using the models of disability.

Methods

As a general guide, you need to provide enough detail in the methods so that someone could replicate the study. Currently, it is not clear how you conducted the research-  More detailed now.

As this research was mixed methods, I anticipated a discussion about how the different methods informed one another/how the different methods were brought together to create a cohesive story about children with mental disabilities in Kenya. More detailed have been included.

Need to step through the different methods you used clearly. There were 3 different approaches (survey, interviews, and focus group) but then I reached section 6 and it was unclear what the purpose of this section was, was this information collected from the research and how it fitted with the research.- This has been clarified.

I find the research questions overly complicated, can it be written more clearly and concisely? This is also the first time I think health is explicitly mentioned (I may be wrong, but up until now, I hadn’t realised health was part of the research question). Health is put under the Medical model of disability.

I would suggest renaming the sub sections of the Methods to something more meaningful (e.g., 5.1 Survey of parents, 5.2 Key informant interviews, 5.3 Parent focus groups). Have chosen to categorise through the models of disability.

You need to include a discussion of the data analysis. Key information missing is how you analysed the qualitative data and how you managed/analysed the different data you collected.Done.

Sections 5.3.1-5.3.4: I’m not sure what the purpose of these sections are? It starts off as describing the participants but then draws on literature and then starts presenting qualitative findings. Need to make it clear to the reader what is happening here. Clarified.

Need to include information about ethical approval and consent. Done.

One the first page, the first author has a footnote about being a parent and bringing two perspectives to the research, this needs to be discussed in the methods. Has been discussed.

Section 6/Results?

From the first few sentences of this section, this looks like a case study about two schools and a sheltered workshop, but how did the researchers come to write these case studies? Was it part of the data collection process? Part of the analytical process? Was this the aim of the study to describe the different settings children with mental disabilities find themselves in which would inform the research question?. The section of schools has been used to explain how the moral model of disability affects caregives and children with developmental disabilities.

I have similar concerns about section 7 and section 8. How do these sections fit in the research? If the purpose was to create case studies, this needs to be discussed in the methods and then signposted in the results when the case studies begin.- Has been clarified.

Discussion

Generally, the discussion should summarise the key findings, discuss how these relate to the literature, limitation, future research/policy recommendations. The discussion in this manuscript would benefit from a clearer structure like this. The discussions and recommendations are clarified.

Make sure to clearly state what this article has added in the discussion and what should the next steps be?

Reviewer 2 Report

Thank you for the opportunity to review "Child Mental Disabilities, Care-Givers Burden and Its Implications on Global Migration: A Case Study of Kiambu County, Kenya". The manuscript, in its current form, has a number of significant problems and as such I cannot recommend it for publication.

The authors have been careless in a number of places (e.g. referencing is incorrect throughout the manuscript) and overall the article lacks structure with no clear introduction, aims/research questions/hypothesis, methods, results, discussion and conclusion. For example information about research questions and hypotheses are provided throughout the methods section when they are better included at the end of the introduction. I suggest the authors seek out additional support from a skilled academic writer in this regard.

I have outlined some specific feedback on the introduction and methods section below, however the results, discussion and conclusion sections all require considerably work.

Introduction:

·         The introduction it is very difficult to read and the study is not properly justified. I suggest carefully editing the introduction to ensure it has a clear structure and the paragraphs flow. For example the aim of the project is mentioned numerous times throughout the introduction in different places and changes each time e.g. Line 59 “This article will demonstrate why there needs to be a societal change in the way child intellectual disability is understood, and handled in developing countries such as Kenya.” Line 71 “For this study, which is funded by the Swiss government, the aim is to look at how mental health of children can be a factor that motivates parents to migrate in search of a better life for their child.” Line 84 “This study is the first part of a project that will eventually look at the situation of migrant families living in Switzerland, in particular whose children suffer from intellectual disabilities.” Line 92 “This article will first give a background on education and child disability in Kenya from a historical point of view etc.” so it is not clear what the actual aim of the study is? Usually it is included at the end of the introduction together with any specific research questions or hypotheses.

·         Regarding terminology, if ‘mental disabilities’ is the preferred terminology in Kenya please define it once clearly and concisely and then continue to this terminology throughout the manuscript - do not use it interchangeably with ‘mental illness’ or ‘mental health’ as these mean different things and readers from other countries may assume it means conditions such as depression, bipolar etc.

Methods:

·         This section is unclear and appears to mix methods and results and also qualitative and quantitative methods. For example qualitative data collection methods are described under the quantitative design heading.

·         It is not clear what questions were asked in the questionnaire, interviews and focus groups.

·         Data analysis methods are poorly described – stating “Inferential statistics was used…” is not clear – what specific statistical tests were used? Furthermore I believe the wrong statistical tests were used.

I wish the authors the best of luck in revising their article.

Author Response

REVIEW 2

Thank you for the opportunity to review "Child Mental Disabilities, Care-Givers Burden and Its Implications on Global Migration: A Case Study of Kiambu County, Kenya". The manuscript, in its current form, has a number of significant problems and as such I cannot recommend it for publication.

The authors have been careless in a number of places (e.g. referencing is incorrect throughout the manuscript) and overall the article lacks structure with no clear introduction, aims/research questions/hypothesis, methods, results, discussion and conclusion. For example information about research questions and hypotheses are provided throughout the methods section when they are better included at the end of the introduction. I suggest the authors seek out additional support from a skilled academic writer in this regard.- Additional editor was sought after.

I have outlined some specific feedback on the introduction and methods section below, however the results, discussion and conclusion sections all require considerably work.

Introduction:

·         The introduction it is very difficult to read and the study is not properly justified. I suggest carefully editing the introduction to ensure it has a clear structure and the paragraphs flow. For example the aim of the project is mentioned numerous times throughout the introduction in different places and changes each time e.g. Line 59 “This article will demonstrate why there needs to be a societal change in the way child intellectual disability is understood, and handled in developing countries such as Kenya.” Line 71 “For this study, which is funded by the Swiss government, the aim is to look at how mental health of children can be a factor that motivates parents to migrate in search of a better life for their child.” Line 84 “This study is the first part of a project that will eventually look at the situation of migrant families living in Switzerland, in particular whose children suffer from intellectual disabilities.” Line 92 “This article will first give a background on education and child disability in Kenya from a historical point of view etc.” so it is not clear what the actual aim of the study is? Usually it is included at the end of the introduction together with any specific research questions or hypotheses.- clarified.

·         Regarding terminology, if ‘mental disabilities’ is the preferred terminology in Kenya please define it once clearly and concisely and then continue to this terminology throughout the manuscript - do not use it interchangeably with ‘mental illness’ or ‘mental health’ as these mean different things and readers from other countries may assume it means conditions such as depression, bipolar etc.- The term used in article is developmental disabilities.

Methods:

·         This section is unclear and appears to mix methods and results and also qualitative and quantitative methods. For example qualitative data collection methods are described under the quantitative design heading.- Has been rectified.

It is not clear what questions were asked in the questionnaire, interviews and focus groups. -  I can attach a questionnaire as supplementary material when submitting final article.

·         Data analysis methods are poorly described – stating “Inferential statistics was used…” is not clear – what specific statistical tests were used? Furthermore I believe the wrong statistical tests were used.- Kindly advise on which tests would have been prefereable.

I wish the authors the best of luck in revising their article. -  Thanks

Reviewer 3 Report

This paper commences with great promise in terms of its title and the study aims/questions which are stated as:"Are barriers to inclusive education and health care reason enough to want to migrate? If yes,what can be done to ensure that all social actors are sensitised about intellectual disabilities? However, I do not feel the paper adequately answered these questions. Whilst the rationale put forward in the introductory section gave some justification for the migration question to be explored, the methodology adopted did not  sufficiently answer these basic questions. Having said that, I believe the content has great promise if it were to be revised and re-submitted

The paper has many strengths, but some weaknesses which I believe can be overcome with careful revision. Suggestions and comments follow which may assist the authors:

I suggest the target population being targeted would be better described as children with intellectual and related developmental disabilities, including children with Down syndrome, autism, epilepsy and cerebral palsy. The authors assert that the term which will be used throughout the paper will be "Mental Disabilities -MD". However, the term which seems to be more consistently used is "intellectual disabilities". Reference to how terms are used in the English speaking world could be briefly explored, especially the two major ones, "intellectual and developmental disabilities" and "learning disabilities" which are used to refer to the same population.

have a concern regarding the authors' understanding of internationally recognised diagnostic characteristics of people with intellectual disabilities. For instance, the two examples given for intellectual disabilities are Down syndrome and cerebral palsy. People with Down syndrome make up a significant proportion of those with intellectual disabilities (approx 33 %), but not all people with cerebral palsy have an intellectual disability. The speech difficulties of this group often lead to a misdiagnosis of intellectual disability.It is possible that the lack of people specialised in the diagnosis of people with disabilities in Kenya has led to this difficulty; a difficulty from a historical perspective that all counties have faced at some time in their development of resources. There are many people with an intellectual disability whose aetiology include Fragile X and many other syndromes, but these do require access to genetic testing facilities. A large proportion of people diagnosed with autism also have an intellectual disability. Epilepsy is also a frequent secondary diagnosis for people with intellectual disabilities. But having said the above, I do recognise that the situation described may be providing a snap shot of the level of development of specialised services for children and adults with mental disorders in the part of Kenya being described.

I have a concern also with the use of the term "suffer" to describe the way people with these disorders experience their impairments. Its use stems from the domination of medical practitioners in the delivery of  diagnostic assessment. It reflects the "disease" concept of mental disorders. The authors might need to reflect upon the international literature concerning the debates regarding the "medical' and "social models" of disabilities.

It appears the Swiss funding agency which supported the study was interested in the migration issue which is well argued in the Background to the study. For instance, it is stated that "This research work goes a step further to determine whether these factors play a role as a migration incentive for parents of such children." As this is the first part of a two-part study, I suggest the aim stated above would be better answered by surveying the parents who chose to migrate in the second part of the research project..

The results documented in this paper do not provide any direct answers to the migration question. Rather, they do proved an excellent snapshot of the education and health services available to the target population in a specific area of Kenya. I suggest that this should be the main focus of a revision.

The theoretical framework is well-argued and provides a sound context for the study.

The mixed method outline is sound, but its application appears to present some difficulties, especially the quantitative aspects. A sampling formula has been presented, but appears it may not have been adopted.For example, it is stated that: "A sample size of 90 was used due to the following factors; first the availability of the time for the study....Secondly, the budget for the study..." This is a legitimate explanation, but it did not need to be substantiated by a needless statistical equation to give it apparent rigour.Another comment was made that only high SES families were chosen, but the survey data includes low SES families as well, although fewer in number (Table 1).

I have reservations about the use of non-parametric analyses of parts of the Survey data, where most of the results were not significant, but no comments were made concerning those results. I suggest these analyses do not strengthen the study. The use of these statistical tests may give a semblance of rigour, but they detract from, rather than enhance the results.

A revision would benefit from a sub-heading Results followed by a more detailed Discussion than is presently provided. The conclusion could segue into the rationale for the next part of the study where the migration issue could be more fully explored.

Author Response

REVIEW 3

This paper commences with great promise in terms of its title and the study aims/questions which are stated as:"Are barriers to inclusive education and health care reason enough to want to migrate? If yes,what can be done to ensure that all social actors are sensitised about intellectual disabilities? However, I do not feel the paper adequately answered these questions. Whilst the rationale put forward in the introductory section gave some justification for the migration question to be explored, the methodology adopted did not  sufficiently answer these basic questions. Having said that, I believe the content has great promise if it were to be revised and re-submitted

The paper has many strengths, but some weaknesses which I believe can be overcome with careful revision. Suggestions and comments follow which may assist the authors:

I suggest the target population being targeted would be better described as children with intellectual and related developmental disabilities, including children with Down syndrome, autism, epilepsy and cerebral palsy. The authors assert that the term which will be used throughout the paper will be "Mental Disabilities -MD". However, the term which seems to be more consistently used is "intellectual disabilities". Reference to how terms are used in the English speaking world could be briefly explored, especially the two major ones, "intellectual and developmental disabilities" and "learning disabilities" which are used to refer to the same population.- The term used throughout the article is developmental disabilities.

I have a concern regarding the authors' understanding of internationally recognised diagnostic characteristics of people with intellectual disabilities. For instance, the two examples given for intellectual disabilities are Down syndrome and cerebral palsy. People with Down syndrome make up a significant proportion of those with intellectual disabilities (approx 33 %), but not all people with cerebral palsy have an intellectual disability. The speech difficulties of this group often lead to a misdiagnosis of intellectual disability.It is possible that the lack of people specialised in the diagnosis of people with disabilities in Kenya has led to this difficulty; a difficulty from a historical perspective that all counties have faced at some time in their development of resources. There are many people with an intellectual disability whose aetiology include Fragile X and many other syndromes, but these do require access to genetic testing facilities. A large proportion of people diagnosed with autism also have an intellectual disability. Epilepsy is also a frequent secondary diagnosis for people with intellectual disabilities. But having said the above, I do recognise that the situation described may be providing a snap shot of the level of development of specialised services for children and adults with mental disorders in the part of Kenya being described.-This has been incorporated in the arguments by referring to the other models of disability.

I have a concern also with the use of the term "suffer" to describe the way people with these disorders experience their impairments. Its use stems from the domination of medical practitioners in the delivery of  diagnostic assessment. It reflects the "disease" concept of mental disorders. The authors might need to reflect upon the international literature concerning the debates regarding the "medical' and "social models" of disabilities.- ‘Suffer’ has been removed.

It appears the Swiss funding agency which supported the study was interested in the migration issue which is well argued in the Background to the study. For instance, it is stated that "This research work goes a step further to determine whether these factors play a role as a migration incentive for parents of such children." As this is the first part of a two-part study, I suggest the aim stated above would be better answered by surveying the parents who chose to migrate in the second part of the research project..- Section on migration is more detailed now.

The results documented in this paper do not provide any direct answers to the migration question. Rather, they do proved an excellent snapshot of the education and health services available to the target population in a specific area of Kenya. I suggest that this should be the main focus of a revision.- Have refocused on migration as main issue.

The theoretical framework is well-argued and provides a sound context for the study. - Thanks

The mixed method outline is sound, but its application appears to present some difficulties, especially the quantitative aspects. A sampling formula has been presented, but appears it may not have been adopted.For example, it is stated that: "A sample size of 90 was used due to the following factors; first the availability of the time for the study....Secondly, the budget for the study..." This is a legitimate explanation, but it did not need to be substantiated by a needless statistical equation to give it apparent rigour.Another comment was made that only high SES families were chosen, but the survey data includes low SES families as well, although fewer in number (Table 1).- Has been clarified.

I have reservations about the use of non-parametric analyses of parts of the Survey data, where most of the results were not significant, but no comments were made concerning those results. I suggest these analyses do not strengthen the study. The use of these statistical tests may give a semblance of rigour, but they detract from, rather than enhance the results.- Are no longer in the article but are now supplementary material.

A revision would benefit from a sub-heading Results followed by a more detailed Discussion than is presently provided. The conclusion could segue into the rationale for the next part of the study where the migration issue could be more fully explored.- Will be put into consideration.

Round 2

Reviewer 1 Report

The revisions have improved the manuscript, however, there are still some areas, particularly a clearer and more detailed methods, an improved migration results section and an expanded discussion to include some future research/implications.

General comments

Well done fixing the in-text citations, however, be sure to revise the manuscript and check for instances where the authors name has been removed (e.g., page 3, line 78 – Already in Switzerland, [18], have documented the link…’)

The manuscript has some formatting errors, capital letters in the middle of sentences, incorrect use of commas, inconsistent use of terms etc. Down syndrome (from my understanding) Down should be capitalised. There are still places in the manuscript that use mental disability or learning disorder – just double check language for consistence.

Revise your use of footnotes, some of the information there is more like a reference or would better placed in the main text

The manuscript has sections that can be edited to make it more concise. For example, irrelevant information or repeated information

Abstract

a great revision, more concise and the results section is much better

Introduction, background and theoretical framework

The discussion around defining disability and developmental disability still needs revision for clarity and to be more concise. Some readers won’t know the difference between developmental disability, intellectual disability, Down syndrome and how these are commonly grouped under developmental disability or how in Kenya a large number of disabilities are grouped under ‘mental disabilities’ as noted in footnote #5.

This section could be referenced more clearly. For example, page 2, line 51 is not referenced and you haven’t introduced the CPRD and if it is ratified in Kenya – this shouldn’t be a footnote either – given the human rights discussion later in the article, this is important

The information about ‘this study’ should be more concise and you need to explain how this study will contribute to the broader study – then you can make connections from the findings of this study in your discussion to the wider study and other areas of future research or policy implications

Page 3, lines 77-82: It’s unclear the link between this evidence and the current study

 I wonder if the ‘this study’ information would be better placed after the theoretical framework. For example, introduction (background information about Africa), disability terminology, models of disability used in Kenya, aim of study, methods…

Page 4, lines 113-119: this discussion around the ICF needs revising to improve clarity

Methods

I think you need a dedicated section for the methods. This will help frame the writing and make the information easier for the reader to find.

You are still missing a discussion around the ethnographic approach you used. I believe with this discussion, you will be able to give the reader a better sense of why you have presented the results the way you have.

One way to improve the methods could be:

Methods: explain the ethnographic approach – ‘This is a mixed method ethnographic study exploring …  Data was collected using four methods (survey, interviews, focus group, participant observation) from a range of key stakeholders (list participants and organisations)’. This would be the place to move the discussion about the positioning of the principle researcher. It would also be a good place to make the connection here about the data analysis and the presentation of the data. For example, ‘the ethnographic approach lent itself to a narrative presentation of the results. We used the data to construct descriptive narratives of the key areas related to children with developmental disability; primary and secondary caregivers, education, healthcare within the frameworks of the different models of disability.’ these are just ideas, but you need to give the reader some indication of how you used the data from the study to form and present the narratives you have. (Please note, narratives might not be the best way to describe these, see what you think!). I think you should make a note that the narratives are supported by quotes and literature where appropriate – you don’t generally find evidence in a results section so you should clarify this for the reader.

You still need a clearer data collection section. I suggest removing the quantitative and qualitative headings and discuss the four methods for data collection. There is still some methods information in the results – move this information here (e.g., page 8, lines 266-274)

 Intertwined within the data collection methods, you should describe the participants as well as the organisations you observed.  

The ethics section doesn’t need the part about human tissue etc. How did you approach the consent of the organisations you observed and the consent from the staff at those organisations?

Remove the study limitations section. This belongs in the discussion

Either in the methods or the ‘this study’ section, you need to make it clear to the read the purpose/aim of this study. After rereading this manuscript, I know that it will inform a broader project in Switzerland, but I don’t understand how. I think the manuscript can be strengthened if you can articulate this. It will also strengthen the discussion if you can indicate how this study will inform other work and be used to impact policy or change.

Page 6, line 176-182 the placement of this section is awkward. This is probably better suited to the introduction/background.

Page 6, line 183-184: you should indicate that each section will include a short contextual paragraph followed by the results

Results

Page 6, line 212: should you be referring to a participant/person by name? Does this breach confidentiality?

Section 5.4 contains a lot of background information that isn’t about Saint Patrick’s school (line 288-295 and 314-318). Make 5.3 and 5.4 short descriptions of the schools and then move on to the results.  

Section 8: Given the focus on this article is about migration (as indicated by the title), these sections should be strengthened. The previous pages have been very detailed about the experiences of children with developmental disability in Kenya so this section feels very ‘light’

Discussion

This section needs to be referenced

Page 17, line 645-648: where did these attributes come from?

As I’ve already mentioned above, the discussion is missing a future research/policy implications section – what has this study added and how can we use this knowledge to improve x,y,z? I’ve gathered this is part of a larger study that does aim to improve something, so it would be worth talking about this here if possible

Author Response

It is uploaded as a PDF.

Reviewer 3 Report

The modifications and additions made in the revised draft have significantly strengthened this paper. I suggest that a final check is needed to make sure the style is consistent with the Journal's requirements. This applies particularly to the way authors are cited in the text. I also suggest the tables cited should be available in the text, and not in a supplement the reader has to find.

I assume the copy editor for the Journal will ensure consistent grammatical presentation. For instance, there is at least one occasion when the text moves from 3rd to 1st person and some sentence constructions may need checking.Keep "Down syndrome" consistent throughout.

I also suggest reference to "autistic child" or "autistic person" is stigmatising. "A child or person with autism" is preferred.

Overall, a much improved presentation of a highly relevant topic.

Author Response

RESPONSE TO REVIEWER 3

The modifications and additions made in the revised draft have significantly strengthened this paper. I suggest that a final check is needed to make sure the style is consistent with the Journal's requirements. This applies particularly to the way authors are cited in the text. I also suggest the tables cited should be available in the text, and not in a supplement the reader has to find. I assume the copy editor (will confirm with journal) for the Journal will ensure consistent grammatical presentation. For instance, there is at least one occasion when the text moves from 3rd to 1st person and some sentence constructions may need checking. Keep "Down syndrome" consistent –Done.

I also suggest reference to "autistic child" or "autistic person" is stigmatising. "A child or person with autism" is preferred. Done.

Overall, a much improved presentation of a highly relevant topic.